# The Neurotrophin System in the Postnatal Brain—An Introduction

**DOI:** 10.3390/biology13080558

**Published:** 2024-07-24

**Authors:** Oliver von Bohlen und Halbach, Monique Klausch

**Affiliations:** Institut für Anatomie und Zellbiologie, Universitätsmedizin Greifswald, Friedrich Loeffler Str. 23c, 17489 Greifswald, Germany; monique.klausch@uni-greifswald.de

**Keywords:** BDNF, aging, learning, memory, obesity, polymorphism

## Abstract

**Simple Summary:**

In this review, the neurotrophin system in the postnatal brain will be introduced. Neurotrophins are proteins that play a crucial role in the development, maintenance, and function of the nervous system. They can bind to specific receptors, including the Trk family of receptors and a low-affinity receptor called p75NTR. These proteins are involved in synaptic and neuronal plasticity, which affects learning, memory, and cognition. Disturbances in the neurotrophin system can contribute to psychiatric diseases. Additionally, age-related changes in brain function correlate with alterations in neurotrophin expression levels.

**Abstract:**

Neurotrophins can bind to and signal through specific receptors that belong to the class of the Trk family of tyrosine protein kinase receptors. In addition, they can bind and signal through a low-affinity receptor, termed p75NTR. Neurotrophins play a crucial role in the development, maintenance, and function of the nervous system in vertebrates, but they also have important functions in the mature nervous system. In particular, they are involved in synaptic and neuronal plasticity. Thus, it is not surprisingly that they are involved in learning, memory and cognition and that disturbance in the neurotrophin system can contribute to psychiatric diseases. The neurotrophin system is sensitive to aging and changes in the expression levels correlate with age-related changes in brain functions. Several polymorphisms in genes coding for the different neurotrophins or neurotrophin receptors have been reported. Based on the importance of the neurotrophins for the central nervous system, it is not surprisingly that several of these polymorphisms are associated with psychiatric diseases. In this review, we will shed light on the functions of neurotrophins in the postnatal brain, especially in processes that are involved in synaptic and neuronal plasticity.

## 1. Neurotrophic Factors

Neurotrophic factors, which are neuropeptides, influence the growth, differentiation, and survival of cells in both the developing and postnatal nervous systems. These factors play essential roles, not only in maintaining neuronal populations but also in facilitating plastic changes, including synaptic and neuronal plasticity. These changes can significantly impact processes related to learning and memory. Aside from the neurotrophins, other protein families belong to the neurotrophic factors, as, e.g., members of the families of ephrins, epidermal growth factors (EGFs), fibroblast growth factors (FGFs), ghrelin, insulin, glial cell line-derived neurotrophic factors (GDNFs), insulin-like growth factors (IGFs), leptin, pituitary adenylate cyclase-activating polypeptide (PACAP), as well as the transforming growth factor-ß (TGF-ß) superfamily and the neurotrophins [1].

## 2. Neurotrophins and Their Receptors

### 2.1. The Neurotrophin Family

In 1951, Rita Levi-Montalcini and Viktor Hamburg published a paper that demonstrated that a tumor released a diffusible factor that promoted neurogenesis [2]. Later, together with Stanley Cohen, they purified a growth-promoting factor [3], which was later termed nerve growth factor (NGF). Since then, several other members of the neurotrophin family have been discovered. Brain-derived neurotrophic factor (BDNF), a further member of the neurotrophin family, was first identified in 1982 as a factor that promoted the survival of cultured embryonic chick sensory neurons [4] and was cloned in 1989 [5]. The gene locus of human BDNF is located on chromosome 11 [6]. The human BDNF has been shown to consist of multiple 5′ noncoding exons and one coding exon, which give rise to alternatively spliced transcripts; moreover, the human BDNF gene has nine functional promoters that are tissue- and brain-region specific [7]. In 1990, a further member of the neurotrophin family, namely neurotrophin 3 (NT-3), was discovered [6]. One year later, neurotrophin-4 (NT-4) was cloned from Xenopus by PCR using primers derived from regions conserved between NGF, BDNF and NT-3 [8]. However, the mammalian NT-4 was much more divergent from its amphibian counterpart than the other members of the neurotrophin family. It differs so much that another group, that had isolated the same gene by using a different approach, called it neurotrophin-5 [9]. Nowadays, this neurotrophin is also named neurotrophin NT-4/5 to denote the mammalian counterpart of the Xenopus NT-4. In addition, there are reports that a further neurotrophin, termed neurotrophin-6, may be expressed by various species and that NT-6 interacts with heparan sulfate proteoglycans [10,11], possibly playing a role in axonal guidance [12].

### 2.2. Receptors for Members of the Neurotrophin Family

The tropomyosin receptor kinases (trks) belong to the family of tyrosine receptor kinase (also termed receptor tyrosine kinases (rtks)), which are high-affinity cell surface receptors. The trk receptors are transmembrane receptor tyrosine kinases, consisting broadly of three domains: an extracellular domain that serves as a binding site for neurotrophins, a transmembrane domain, and an intracellular domain responsible for tyrosine kinase activity [13]. Currently, three different Trk receptors have been identified to which the neurotrophins bind: NGF binds specifically to trkA, and BDNF and NT-4 bind to trkB. NT-3 mainly activates trkC but can also bind with lower affinity to trkA and trkB in certain cell systems [13,14]. In addition, all neurotrophins can signal through the low affinity receptor p75 neurotrophin receptor (p75NTR) [13,15].

The tropomyosin receptor kinase A (trkA; also known as high affinity nerve growth factor receptor, neurotrophic tyrosine kinase receptor type 1, or trk1-transforming tyrosine kinase protein) is encoded in humans by the NTRK1 gene. The NTRK1 gene is a member of a receptor family of tyrosine kinases, which also include trkB and trkC. The binding of NGF to trkA leads to a ligand-induced dimerization. After complex formation, the NGF/TrkA complex is transported by endocytosis from the synapse to the cell body, where it then activates NGF-dependent transcription programs. After activation, the tyrosine residues are phosphorylated, which recruit signal molecules that activate various pathways that can lead to neuronal differentiation and survival of neurons. Two pathways through which this complex acts in promoting growth are the Ras/MAPK pathway and the PI3K/Akt pathway [16].

TrkB is encoded by the NTRK2 gene and is a further member of a receptor family of tyrosine kinases. Binding of BDNF to trkB receptors induces ligand–receptor dimerization and auto-phosphorylation of tyrosine residues. Three main intracellular signaling cascades are activated by trkB receptors [17]: (i) the Ras pathway, (ii) the PI3K-Akt pathway and (iii) the PLCgamma—Ca^2+^ pathway. Currently, several trkB isoforms in the mammalian central nervous are known [13,18]: (i) a full-length isoform (TK+) that represents the typical tyrosine kinase receptor that transduces the BDNF signal via the above-mentioned signaling pathways and (ii) truncated isoforms (TK−) that are composed of the same extracellular domain and transmembrane domain as TK+. However, they have very short cytoplasmic domains and lack the entire kinase catalytic region.

TrkC is the high affinity receptor for NT-3. Similar to trkA and trkB, binding of the neurotrophin induces receptor dimerization followed by trans-autophosphorylation of the intracellular domain of the receptor. This event triggers downstream signaling cascades that are involved in the regulation of diverse functions, such as, e.g., cell survival and proliferation. Somewhat comparable to trkB, trkC also exists in different isoforms, e.g., lacking the kinase domain [13,19,20]. Currently, the functions of these truncated trkB and trkC isoforms are poorly understood and it is thought that the truncated receptors alone can affect intracellular signaling. The truncated isoforms may be capable of raising local neurotrophin concentration by capturing and presenting neurotrophins to neurons expressing full-length trk receptors. Truncated receptors may also inhibit the activation of Trk kinases by forming non-productive heterodimers [16,20]. The signaling through trk receptors is quite diverse, not only since full-length and truncated isoforms do exist, but also due to the fact that the sorting receptor sortilin can interact with trk receptors and thus enables their anterograde axonal transport, thereby enhancing the neurotrophin signaling [21].

A further receptor, which is able to function as a neurotrophin receptor (NTR), is a member of the tumor necrosis factor receptors and has a molecular weight of 75 kDa; based on this, the name p75NTR is derived for this receptor. P75NTR is also a transmembrane receptor, but in contrast to the trk receptors it binds all neurotrophins with relatively low affinity [22,23]. This receptor is also known as low-affinity nerve growth factor receptor (LNGFR). The p75NTR receptor can be found in the adult forebrain primarily on cholinergic neurons (both projection neurons and interneurons), which are located particularly in the septal area, the nucleus basalis Meynert, the nucleus caudatus, and the putamen [24]. The functions of p75NTR are quite diverse. Many of its modes of action result from its function as a co-factor for other receptors, and some are independent of other receptors [25]. Within the receptor exists a so-called “death domain” that serves regulated cell death as well as other apoptosis-initiating functions [23,26]. Examples of these apoptosis-initiating functions include activation of the JNK-p53-Bax signaling cascade, as well as regulation of numerous proteins that initiate apoptotic processes. Among these proteins are NADE (p75-associated cell death executor), NRAGE (neurotrophin receptor-interacting melanoma antigen-encoding protein), NRIF (neurotrophin receptor-interacting factor), and SC-1 (Schwann-cell factor 1), which initiate apoptosis via cell cycle intervention or caspases [14,23]. On the other hand, p75NTR can also promote neuronal survival via activation of the transcription factor “nuclear factor κB”, also known as NFκB [14]. Roughly spoken, (full-length) trk receptors can transmit positive signals, such as enhanced survival and growth, while the receptor p75NTR is capable of transmitting both, positive and negative signals. The signals generated by the two neurotrophin receptors can either augment or oppose each other [25].

## 3. Polymorphisms in Genes Coding for Neurotrophins

In humans, several polymorphisms in the genes coding for the different neurotrophin receptors have been reported. For example, it is known that human NTRK1 gene that encodes trkA is responsible for congenital insensitivity to pain with anhidrosis (CIPA), an autosomal recessive disorder characterized by a lack of pain sensation and anhidrosis [27,28]. Concerning the human NTRK2 gene that encodes trkB, it has been reported that one polymorphism in that gene was associated with major depression and the rate of suicidal attempts [29]. There is also evidence that polymorphisms in the human NTRK3 gene, coding for trkC, are involved in mood disorders [30], bipolar disorders [31] and schizophrenia [32]. Concerning p75NTR, there are several publications reporting an association between p75NTR polymorphisms and major depressive disorders [33,34,35]. Several of these polymorphisms are related to dysfunctions of the central nervous system. Thus, polymorphisms in the genes coding for the neurotrophins might also be related to dysfunctions of the central nervous system and might be associated with mental illnesses or with alteration of neuronal plasticity.

In humans, several polymorphisms in the genes coding for different neurotrophin have also been reported. Concerning NT-3, for example, associations of neurotrophin-3 (NT-3) gene polymorphisms with schizophrenia have been reported by two different groups [36,37], whereas in another study no association was found [38]. Furthermore, there is evidence that polymorphisms in NGF may contribute to schizophrenia susceptibility and psychopathology [39]. However, most polymorphisms in a gene coding for a neurotrophin that are associated with a specific phenotype are related to the human BDNF gene (Figure 1). Indeed, related to BDNF, there are several known single nucleotide polymorphisms. The most common single nucleotide polymorphism in the BDNF gene is rs6265. This point mutation is located in the coding sequence. At position 196, a guanine to adenine switch occurs that results in a switch of the amino acid. Thus, a valine to methionine exchange at codon 66, Val66Met, occurs [40]. Polymorphisms in the human BDNF gene have been shown to alter memory functions and the functioning of the hippocampal formation [41,42]. In detail, the BDNF Val66Met polymorphism has an impact on the activity-dependent secretion of BDNF in synapses, that is crucial for learning and memory [43] and deficit in memory performance has been noted in individuals homozygous or heterozygous for the met allele [43]. Aside from deficits in long-term memory, patients carrying the BDNF val66met polymorphism display deficits in working memory performance [44]. BDNF polymorphisms are not only linked to poorer working memory performance but also to differences in the prefrontal cortex and reduced hippocampal volumes [45]. These anatomical changes might hint that BDNF not only plays a role in working memory, but in other memory processes at well. The results of a study by Cao and colleagues pointed in this direction, since they discovered that bipolar disorder patients carrying the BDNF val66met met allele not only displayed deficits in memory performance, but also have a reduced volume of the hippocampus [46]. Since the hippocampal formation plays an important role in learning and memory and reductions in the size of the hippocampal formation might be responsible for memory deficits, the hippocampal formation might play a very prominent role in in the memory deficits seen in the BDNF val66met polymorphism. Thus, the BDNF Val66Met polymorphism may represent a risk factor associated with cognitive impairment [47] or multiple disturbances in memory systems [43]. Genetic association studies have also identified different loci of genetic variation within the BDNF gene that may be associated with both anxiogenic- and anxiolytic-like effects [48]. Furthermore, BDNF polymorphisms have been found to be associated with major depression, bipolar disorders, schizophrenia and eating disorders (for more details see the excellent review by [49]). Several recent studies indicate that BDNF polymorphisms are associated with obesity [50,51]. However, in another study no association of a BDNF polymorphism with increased body mass index or major depression was seen [52].

Aging is associated with cognitive decline and increased risk of neurodegenerative diseases, such as Alzheimer’s disease (AD) and Parkinson’s disease (PD). Several studies have shown that BDNF levels decrease with age in humans and animals [53,54]. This decline may contribute to the impairment of synaptic plasticity and neuronal function, leading to cognitive deficits and neurodegeneration. Thus, it is not surprisingly that BDNF Val66Met polymorphism affects aging in multiple types of memory [43] and that BDNF polymorphisms play diverse roles in the pathophysiology of aging-related diseases [55] as, e.g., PD [56] or AD [57]. Interestingly, DAT1 (dopamine transporter 1, also known as SLC6A3) and BDNF polymorphisms interact to predict Abeta and tau pathology, as well as hippocampal atrophy [58].

## 4. Adult Hippocampal Neurogenesis, Voluntary Exercise and Major Depression

The neurotrophins, and especially BDNF, are key regulators of different neurotransmitter systems in the brain. Thus, BDNF, acting through trkB receptors, is known to enhance growth and survival of serotonergic neurons [59,60]. In the adult rat brain, BDNF is able to promote regenerative sprouting of serotonergic fibers, but not the survival of injured serotonergic axons [61]. Neurotrophins are also involved in the regulation of the dopaminergic system. For example, BDNF is known as a neurotrophic factor for the dopaminergic neurons located in the substantia nigra [62,63]. BDNF is able to promote the survival of cholinergic neurons in the forebrain and can protect them from dying after injuries caused by excitotoxic lesions [64]. Thus, neurotrophins play essential roles in the regulation of brain function. Alterations in the neurotrophin system also affect the wiring of brain areas that are involved in learning and memory. For example, mice with reduced levels of BDNF [65] or trkB [66] display reduced numbers of dendritic spines.

Dendritic spines are small postsynaptic structures that play a pivotal role in neuronal plasticity, learning and memory under physiological as well as pathophysiological conditions [67]. Changes in the morphology or densities of dendritic spines are thought to be fundamental for neuronal signaling. As mentioned above, mice with reduced levels of BDNF or trk B display disturbed dendritic spine densities. As already mentioned, the human Val66Met polymorphism in the BDNF gene is associated with mental illnesses and disturbances (Figure 1). Recently, by using super-resolution imaging, it could be demonstrated that the BDNF Met pro-domain disassembles dendritic spines and eliminates synapses in hippocampal neurons [68]. This elimination might contribute to defects in neuronal plasticity that can translate into behavioral malfunctions, as seen in learning and memory. During sleep, memory consolidation involves interactions between the hippocampus and cortex, in that neural patterns stored in the hippocampus related to these memories are replayed for the cortex and stored within the cortex. In this context, whether BDNF might contribute to this process was analyzed. Carriers of Val66Met polymorphisms, as compared with Val66 homozygotes, showed stronger forgetting overnight (24 h after encoding), but not over shorter time (e.g., 20 min after word list presentation) and no effect of Val66Met genotype on motor learning was observed [69]. From these data, the authors of the study concluded that BDNF plays a role in neuroplasticity underlying episodic memory consolidation during sleep. This may hint at a severe disturbance in memory storage. Along this line, it has been reported that children with learning disorders might display reduced levels of BDNF or might be carriers of the Val66Met polymorphism. The results of a study indicate that there is indeed an association between the Val66Met polymorphism and learning deficits, at least in children between 6.5 and 12 years in an Egyptian population [70].

Within the hippocampus, learning and memory formation can induce the formation of new neurons in the dentate gyrus. This specific generation of new neurons within the postnatal brain is termed “adult neurogenesis”. Adult neurogenesis can be altered through external stimuli and it can be regulated by internal stimuli such as growth factors, including members of the neurotrophin family [71]. Mice heterozygous for BDNF have been reported to display reduced rates of newly formed neurons in the hippocampus as compared to BDNF +/+ mice [72] and intra-hippocampal BDNF infusion has been reported to increase hippocampal adult neurogenesis in rats [73]. In addition, exogenously applied NGF not only increases hippocampal cholinergic activity, but also promotes hippocampal neurogenesis [74].

In addition, anti-depressant treatment has positive effects on adult hippocampal neurogenesis [75], as well as “enriched environment”. An enriched environment is an environment that provides a variety of physical and social stimuli to an organism, such as toys, puzzles, and social interaction. This type of environment has positive effects on the brain including formation of new synapses [76] and, concerning the postnatal hippocampus, in addition the generation of new neurons [77]. Moreover, different forms of exercise can also have positive effects on adult hippocampal neurogenesis. May NGF also play a role in adult hippocampal neurogenesis? At least one recent study may hint at this direction, since adult hippocampal neurogenesis was found to be increased in APP/PS1 mice that were injected with anti-proNGF antibodies [78]. A further indirect link is provided in another study by Darwish and coworkers. They have shown that urinary tract infection is associated with decreased mRNA expression of BDNF, NGF, and FGF2 in the hippocampus, as well as reduced adult hippocampal neurogenesis [79]. It is possible that the effects of NGF on adult hippocampal neurogenesis are mediated through trkA. Rats that underwent whole-brain irradiation display a decrease of more than 70% in the number of doublecortin positive cells [80]. The authors of this study also showed that an overexpression of trkA in the hippocampus not only prevented memory deficit in the irradiated rats but also rescued adult hippocampal neurogenesis [80]. However, the effect of trkA might be of an indirect nature, since the physiological expression of trkA (like p75^NTR^; see e.g., https://mouse.brain-map.org/experiment/show/79591691 accessed on 3 July 2024) in the postnatal hippocampus is rather low, as can be seen, e.g., in the Allen Brain Atlas (https://mouse.brain-map.org/experiment/show/71670679 accessed on 3 July 2024).

Under normal conditions, NT-3 is mainly expressed in the dentate gyrus [81]. Deletion of NT3 in the brain has an effect upon adult hippocampal neurogenesis—not on the proliferation, but on the differentiation of the newly formed cells [81].

Physical exercise, like wheel running, can be induced in rodents by ad libitum access to running wheels. This physical activity can induce, within a short time frame, significant levels of the neurotrophins NGF and BDNF in the hippocampus, as well as in some cortical areas [82,83]. NT-3, however, seems not to be upregulated by voluntary wheel running [84]. Later, it could be shown that physical activity not only elevated BDNF levels in rodents, but also increases circulating BDNF levels, as measured from arterial and internal jugular venous human blood samples [85]. The authors of the study hypothesized that endurance training would enhance the release of BDNF from the human brain, based on the observation that a comparable effect has been described for rodents. Moreover, it has been described that physical activity has beneficial effects on memory functions and adult neurogenesis [86,87]. It thus might be possible that the running-induced increase in BDNF levels, especially in the hippocampus, might be beneficial for adult hippocampal neurogenesis and spinogenesis and, thus, also for hippocampus dependent forms of learning and memory (Figure 2).

Major depression, also known as major depressive disorder (MDD), is a mental health condition characterized by a period of pervasive low mood, low self-esteem, and loss of interest or pleasure. If depression is not addressed, it tends to escalate, leading to a range of emotional, behavioral, and health issues. It can even induce feelings of suicide, attempts at suicide, or actual suicide. Major depression is often associated with reduced hippocampal volume [88,89]. Treatment options for major depression include medication and psychotherapy. Anti-depressant treatment has been shown to affect the neurotrophin system and many studies hint at a role of the neurotrophin system in depression. Thus, treatment with antidepressants can elevate BDNF mRNA levels in the rat hippocampus, as well as trkB mRNA [90,91]. Stress can often precipitate the onset of depression and stress has been shown to decrease BDNF mRNA levels (see for review: [92]). Furthermore, several studies showed that animals receiving BDNF infusions displayed behaviors that are normally associated with antidepressant drugs (see for review: [92]). Thus, the neurotrophin system and especially BDNF seem to be involved in the regulation of pathways that are altered in major depression. Therefore, (antidepressant) treatment leading to an increase in brain BDNF levels might be helpful in the treatment of that mental disorder.

Voluntary physical activity can influence brain plasticity by facilitating neurogenerative, neuroadaptive, and neuroprotective processes. Voluntary exercise has been shown to facilitate depressive symptoms. For example, it has been shown in mice that voluntary exercise produces antidepressant and anxiolytic behavioral effects [93]. In a rat model of depression, it has been shown further that exercise improves not only depressive symptoms, but also increases the numbers of excitatory synapses in the hippocampus [94] and promotes newborn neuron maturation in the dentate gyrus [95]. As mentioned above, voluntary exercise is capable of increasing brain BDNF levels. Thus, there is strong evidence that at least BDNF is associated with increased risk for depression, whereas increasing BDNF by exercise appears to improve memory function and to reduce depression. There are also several lines of evidence that link BDNF to the hippocampal atrophy seen in depression, and to the efficacy of antidepressants (see for review: [96]). However, it is still a matter of debate whether BDNF is a causal factor leading to depression or a correlate of depression. Research that helps to obtain insight in this issue is required since brain BDNF levels might have an impact on suicidal behavior. Suicide is most common in mood disorders, such as major depressive disorder and bipolar disorder [97]. A post-mortem analysis of brains of suicide (and drug-free) victims indicated a significant decrease in BDNF and NT-3 levels in the hippocampus compared with non-suicide controls [98]. Another post-mortem study was performed on the hippocampus of suicide victims compared to non-psychiatric control individuals. The authors of that study reported reduced expression of the neurotrophins BDNF, NGF, as well as of the receptors TrkA and TrkB, on protein level, as well as on the mRNA levels, of postmortem brains of suicide victims [99]. In another study, significantly lower BDNF plasma levels in major depressive disorder (MDD) patients who had recently attempted suicide could be shown compared to non-suicidal MDD patients and normal controls [100].

## 5. The Neurotrophin System and Neuronal Plasticity

Reduced levels of neurotrophins have profound effects on mental health. To obtain more insight in this, it is crucial to understand the functions of the neurotrophin system under normal (healthy) conditions. Neurotrophins play an important role in neuronal plasticity and in brain functions such as, e.g., learning and memory. The generation of different mouse models with deficiencies either in a member of the neurotrophin family or in one of their receptors helped to obtain insight into the diverse role of the neurotrophins in neuronal plasticity. The neurotrophins function to prevent death of embryonic neurons during development [101] and are also crucially involved in the pre-natal and early postnatal development of other organs. These are reasons why a complete deletion of one neurotrophin mainly leads to early postnatal death of the homozygous knockout mice. NT-3 deficient mice displayed severe movement defects (accompanied by loss of cranial and spinal peripheral sensory and sympathetic neurons) and mostly died shortly after birth [102]. Mice lacking NT-4 are viable, while mice with a homozygous deletion of BDNF or NGF die shortly after birth [103,104,105]. Mice with a complete disruption of trkA die early after birth [105], as well as mice with targeted disruption of either trkB [106] or trkC [105]. In contrast to this, p75NTR deficient mice were viable and fertile [107]. However, mice with a heterozygous deficiency for BDNF, NGF, trkA, trkB or trkC (or mice with a conditional deletion of a neurotrophin in a certain cell type) survive into adulthood. Thus, these mouse models were helpful for investigations concerning the roles of neurotrophins in the postnatal brain.

Since the neurotrophins are neurotrophic factors for cholinergic, dopaminergic, as well as serotonergic neurons (see above), reductions in neurotrophin levels (or their receptors) can have an impact on the innervation pattern of the forebrain that persists into adulthood. For example, (heterozygous) deletion of trkB decreases the catecholaminergic innervation densities of some brain areas in adult and aged mice [108].

Deletion of p75NTR increases the cholinergic innervation in the adult brain [109,110,111,112,113]. Interestingly, when in senescence-accelerated mouse prone number 8 (SAMP8; a mouse model of accelerated aging) the p75NTR is deleted, a higher number of cholinergic neurons can be seen at two months of age. However, at an age of 10 months, the number of cholinergic neurons declines [114]. Comparable to the results obtained by analyzing p75NTR deficient mice, NGF-deficient mice also display impaired spatial learning and memory and decreased adult hippocampal volume [115], whereas they display a decreased cholinergic innervation [115]. Interestingly, mice with an increased expression of trkA show increased levels of choline acetyltransferase in forebrain cholinergic regions and their projection areas, such as e.g., the hippocampus [116].

Reduced availability of neurotrophins or their receptors not only affected the innervation pattern of transmitter-releasing neurons but can also alter dendritic structures. Mice deficient for p75NTR display increased dendritic complexities and increased spine densities [117], whereas trkB deficient mice display reduced spine densities [66]. Furthermore, p75NTR seems to be involved in regulating adult hippocampal neurogenesis [118]. Moreover, mice lacking the trkB in hippocampal neurons display impaired neurogenesis [119]. In sum, these studies argue for important functions of the neurotrophins in the adult brain for the maintenance of neuronal networks. Since adult hippocampal neurogenesis has an impact on neuronal plasticity, and since dendritic spines are structures that play fundamental roles in structural and functional plasticity, neurotrophins and their receptors may not only contribute to the maintenance of the postnatal brain but may also be involved in plastic changes. In the postnatal brain, neurotrophins are expressed by cells and, in these cells, the neurotrophins can be transported anterograde or retrograde to other regions and either stored in vesicles or immediately released. For example, BDNF and its pro-peptide can be stored in presynaptic dense core vesicles in neurons [120]. This may argue for dynamic roles of the neurotrophins in neuronal signaling.

Long-term potentiation (LTP) is an electrophysiological correlate of neuronal plasticity and was first described in the early 1970s [121]. LTP is a phenomenon observed in neurons, where the synaptic strength increases persistently after high-frequency stimulation. LTP is used to study changes in synaptic connections that are involved in plasticity, learning, and memory, e.g., in the hippocampal formation [122]. BDNF knockout mice display reductions in hippocampal LTP, suggesting that BDNF might be involved in LTP induction [123]. This effect seems to be mediated by BDNF-TrkB signaling, since trkB knockout mice also display reduced LTP [124]. In addition, deficits in LTP seen in BDNF knockout mice can be rescued either by recombinant BDNF [125] or by virus-mediated BDNF-gene transfer [126]. Interestingly in this context is the observation that transient application of BDNF and NT-3, induces a long-lasting increase of synaptic transmission, likely mediated by trk receptors [127]. Several years later, it was reported that BDNF in the intact adult hippocampus induces LTP that involves phosphorylation of ERK (extracellular signal-regulated protein kinase) and the selective induction of the immediate early gene Arc (activity-regulated cytoskeleton-associated protein) [128]. BDNF can be released during or shortly after LTP induction and might play a critical role as an active instructor of this form of synaptic plasticity, and even BDNF exerts long-lasting effects to maintain LTP (see for review: [129]). The role of BDNF in the regulation of hippocampal LTP has been clearly shown and there is evidence suggesting that BDNF is released from both pre- and postsynaptic sites. High-frequency stimulation of glutamatergic synapses triggers the release of BDNF from synaptically localized secretory granules, a process dependent on postsynaptic ionotropic glutamate receptors and Ca^2+^ influx [130]. This release can occur from extra-synaptic dendritic vesicle clusters, indicating a potential spatial restriction to specific synaptic sites [130]. In the dorsal striatum, presynaptic BDNF promotes postsynaptic LTP, suggesting a role for both pre- and postsynaptic BDNF in synaptic plasticity [131]. However, BDNF is selectively required for the presynaptic component of LTP at hippocampal CA1–CA3 synapses, with its release from presynaptic neurons being crucial for this form of plasticity [132]. These findings highlight the complex and dynamic role of BDNF in synaptic plasticity, with its release from both pre- and postsynaptic sites contributing to different forms of LTP. BDNF has also been shown to modulate LTP through both presynaptic and postsynaptic mechanisms. For example there are reports suggesting a postsynaptic role for BDNF in LTP [133,134] specifically identifying dendritic spines as the site of BDNF action [133]. However, others provide evidence for a presynaptic role [132,135]. These findings suggest that BDNF may act through both presynaptic and postsynaptic mechanisms to modulate LTP (For more details concerning the pre- and postsynaptic source or side of action, see the reviews by [136] and by [129]). It is well established that neurotrophins, especially BDNF, have crucial roles in learning and memory. BDNF is involved in memory formation [137], persistence and storage of memory [138]. Since BDNF was found to be involved in the regulation of activity-dependent plasticity in the hippocampus, which is a key region for learning and memory, down-regulation of the levels of available and functional BDNF are likely to impair mechanisms related to learning and memory. Comparable to mice with a conditional deletion of trkB [124], mice with a conditional deletion of BDNF in the forebrain display learning deficits [139]. Likewise, hippocampus-specific deletion of BDNF in adult mice impairs spatial memory and extinction of aversive memories [140]. Based on this, it is not surprisingly that polymorphisms in the human BDNF gene have been shown to alter memory functions and the functioning of the hippocampal formation. Fewer data are available concerning the other members of the neurotrophin family. It is known that mice with a targeted deletion of neurotrophin-4 gene display deficits in long-term potentiation [141]. Furthermore, it is known that NGF augmentation facilitates induction of LTP, whereas NGF blockade inhibits LTP and impairs spatial memory [142]. A widely used animal model of cholinergic deficit is based on the application of the immunotoxin 192IgG-saporin. Such an application results in a loss of cholinergic neurons and induces a strong reduction of the cholinergic innervation of forebrain areas, including the hippocampus. By using such a model, it could be shown that the magnitude of LTP in the hippocampus is reduced, but that subsequent NGF overexpression reversed this effect [143].

## 6. The Neurotrophin System and Obesity

According to different studies, the prevalence of obesity increases up to 69 years in both men and women and newer studies hint that the prevalence of obesity tends to increase even in the most advanced ages [144]. This may be linked to the BDNF-system, since it has also been observed that during aging there is a decline in BDNF levels [145]. Moreover, in humans, the BDNF Val66Met polymorphism has also been associated with eating disorders and obesity [146,147]. Interestingly, heterozygous BDNF knockout mice gain more weight than control mice and several of the heterozygous BDNF-deficient mice become obese [148], but infusion with BDNF was observed to transiently reverse the eating behavior and obesity [148]. This phenomenon is not related to the reduced levels of BDNF in the periphery but seems to depend on the available BDNF in the brain, since mice with conditional deletions of BDNF in the brain display comparable phenotypes [149,150]. The BDNF-deficient mice develop mature-onset obesity, primarily due to overeating [151]. This effect may relate to the hypothalamus, a brain region involved in the regulation of the eating behavior. In the hypothalamus, BDNF is strongly expressed mainly in neurons of the ventromedial hypothalamus (VMH). BDNF expression in the VMH can be stimulated by signals of nutrient availability or inhibited by nutrient restriction leading to excessive feeding (hyperphagia) and obesity in mice (see for review [152]). In addition, it has been shown that recombinant adeno-associated virus (AAV)-mediated hypothalamic BDNF gene transfer alleviates obesity and that BDNF gene transfer prevents aging-associated weight gain and improves glucose tolerance and suppresses inflammatory genes in the hypothalamus and adipose tissues [153].

Comparable to human BDNF Val66Met carrier, genetic Val66Met BDNF variant increases hyperphagia on high-fat diets (HFD) in mice [154]. Using this mouse model, the results hinted that satiety factors generated during HFD feeding induce BDNF release in order to suppress excess dietary fat intake [154]. In addition, it is likely that BDNF acts downstream of many anorexigenic factors to control body weight and may modulate synaptic plasticity in the hypothalamus, thereby playing a crucial role in the control of energy homeostasis (see for a detailed review on this topic [155]).

Leptin-deficient mice (so-called ob/ob mice) are characterized by a strong weight gain (accompanied by behavioral deficits and changes in adult hippocampal neurogenesis) and represent an animal model of obesity [156]. Measurements of NGF in several tissues of such mice showed a significantly lower concentration in the salivary glands as compared to controls. However, no differences in NGF levels in brown fat or in the cortex, cerebellum, or brain stem were detected in these mice [157]. This may indicate, that at least in this mouse model of obesity, NGF in the postnatal brain does not have a major impact. However, in another recent study, which examined the effects of genistein (a naturally occurring phytoestrogen), it was observed that genistein reduced weight gain in ob/ob mice and increased the levels of NGF and BDNF in whole brain lysates in genistein-treated ob/ob mice [158]. Whether or not brain NFG plays a pivotal role in obesity is not yet clear. However, peripheral NGF may have an impact on body weight. By using an adipocyte-tropic adeno-associated virus (AAV) to deliver BDNF or NGF to subcutaneous white adipose tissue, the effects of BDNF and NGF on adipose tissue in combination to peripheral neuropathy was examined [159]. BDNF application was found to be effective in the beginning, but then the effects decline, whereas NGF application was more effective in the long term concerning tissue innervation; however, the body weight of the different groups of mice was largely unaffected [159]. Thus, neither peripheral nor brain NGF seems to be involved in the regulation of bodyweight.

To date, it is not known whether brain NT-3 or NT4/5 have an impact on the regulation of body weight. In a study from 2018, the effects of training of overweight persons was analyzed and, in this study, it could be shown that the different forms of training are capable of increasing BDNF, NT-3, and NT4/5 concentrations in the blood [160]. Interestingly, in this context, another group analyzed another mouse model of obesity (i.e., high-fat induced obesity). These mice become obese within several weeks. In the study by Bae and coworkers, these mice underwent different forms of training (from mild to high-intensity treadmill training). It could be shown that training was successful in reducing fat mass and increasing the levels of BDNF, NGF and NT-3 in the hippocampus [161]. However, it is possible that the changes in the neurotrophin levels were unrelated to the obesity phenotype and the subsequent training-induced loss of body weight, since it is known that exercise has an impact on adult hippocampal neurogenesis and that neurotrophins play crucial roles in regulating adult hippocampal neurogenesis.

## 7. The Neurotrophin System and Aging

As already mentioned, aging is associated with cognitive decline and reductions in available BDNF. However, the mechanisms underlying the age-related decrease of BDNF are not yet fully understood, but may involve cellular and metabolic changes that result in declines in neuronal plasticity, resulting in a reduction in the number of synapses and plasticity-related proteins [162]. The hippocampus is involved in many aspects of learning and memory and therefore it is not surprising that the hippocampal formation is sensitive to aging. Indeed, aging is accompanied by declines in hippocampal memory performance [163]. It is known that, during normal aging, the hippocampal volume shrinks without a significant loss of neurons in the hippocampal formation; instead, a decline in dendritic spine densities, as well as the in rate of adult hippocampal neurogenesis, can be observed [54]. In humans, increasing age was found to be associated with lower BDNF serum levels [164] and hippocampal volume, as well as lower spatial memory performance [145]. However, since BDNF is produced and secreted by peripheral tissues, changes in serum BDNF might not reflect central changes in BDNF. Webster and colleagues, however, reported that within the human hippocampus the levels of BDNF mRNA did not change significantly with age, but that the levels of trkB mRNA decrease over the life span [165]. In contrast, in the human temporal cortex, BDNF and trkB mRNA levels were highest in neonates and decline with age [165]. In 2020, the proteomic changes in the hippocampus and cerebral cortex of young (4-month) and aged (16-month) mice were analyzed and about 390 hippocampal and 260 cortical proteins were found to be altered (either decreasing or increasing) in the context of aging, whereby BDNF was found to be significantly decreased in the aged hippocampus and cerebral cortex [166]. Thus, at least concerning the hippocampal formation, an age-related decline in BDNF levels exists, which is associated with poorer spatial memory performance and hippocampal shrinkage (Figure 3). However, in the hippocampus, there is not only an age-related decline in BDNF, but also in NGF [167].

## 8. BDNF, Physical Activity and Aging

With advancing age a remarkable decrease in hippocampal BDNF levels could be observed [96,168]. An unlimited access to running wheels in aged mice for three months could rescue this effect and upregulate levels of hippocampal BDNF mRNA compared to standard-housed animals [169]. Furthermore, a higher amount of cell proliferation in the DG could be observed after running exercises [96,170], which also led to improved learning rates in behavioral experiments, like the Morris water maze [96]. It is known that (running) exercise led to a loss of body weights in aged mice of both sexes [169] and influenced physiological and morphological properties of the aged (and young) hippocampus by increasing BDNF expression in the hippocampal formation and cerebral cortex in a dose-dependent manner [96,171]. Nevertheless, increased BDNF levels in plasma and serum mediated by endurance training with moderate intensity could be observed in humans [96,172]. However, the strength of training does not have an impact on BDNF serum levels [173]. Possible explanations are, among others, differences in sex [169], the duration and intensity of exercise, the methods and the timing of BDNF assessment, and the species analyzed. However, accumulating research shows that physical activity reinstates hippocampal function by enhancing the expression of BDNF, which promotes adult neurogenesis, angiogenesis, and synaptic plasticity, as well as counteracting age-associated declines in mitochondrial and immune system function [174]. Physical activity has been shown to improve cognitive performance and reduce the risk of dementia and depressive symptoms in older people (for review, see [175]) Cognitive stimulation, for example through an enriched environment that promotes motor and sensory stimulation, is another positive exogenous factor. Both cognitive stimulation and physical activity lead to an increase in BDNF levels in the brain; this is considered to be the reason why these two exogenous factors have a positive influence on learning and memory processes. Thus, BDNF might represent one of the key molecules that is critically involved in the regulation and maintenance of neuronal plasticity in the adult as well in the aged brain.

## 9. Conclusions

Neurotrophins are neurotrophic factors and the most well-known and best-studied member is BDNF. BDNF is a key molecule for memory formation and maintenance in the healthy and pathological brain. Enriched environment or physical activity can modulate brain neurotrophin levels and neuronal functions, and thus improve cognitive performance and brain health.

BDNF expression and function is affected during aging in various ways, which may contribute to cognitive decline and neurodegeneration. Thus, BDNF may represent a key factor involved in the regulation of functions attributed to neuronal plasticity in the postnatal brain, from the early development phase to old age. Nevertheless, further research is needed to elucidate the optimal dose, intensity, frequency, duration, as well as the type of physical activity for enhancing BDNF levels and functions in different populations. The roles and functions of the other members of the neurotrophin family in the maintenance of the postnatal brain or their role in neuronal plasticity are still far from being understood. Likewise, the possible involvement of neurotrophin during aging as well as in age-related neurodegenerative diseases needs further investigations. This Special Issue on the *Roles and Functions of Neurotrophins and their Receptors in the Brain* will add new and important findings.

## Figures and Tables

**Figure 1 biology-13-00558-f001:**
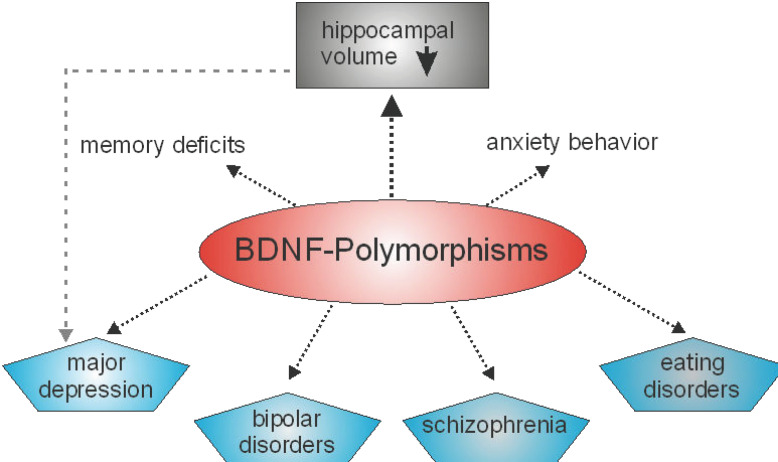
BDNF polymorphisms are associated with different mental illnesses and disturbances in memory systems as well as in anxiety related behavior. Moreover, BDNF polymorphisms can be associated with reductions in the volume of the hippocampus. However, hippocampal volume reduction is also observed in major depression [49].

**Figure 2 biology-13-00558-f002:**
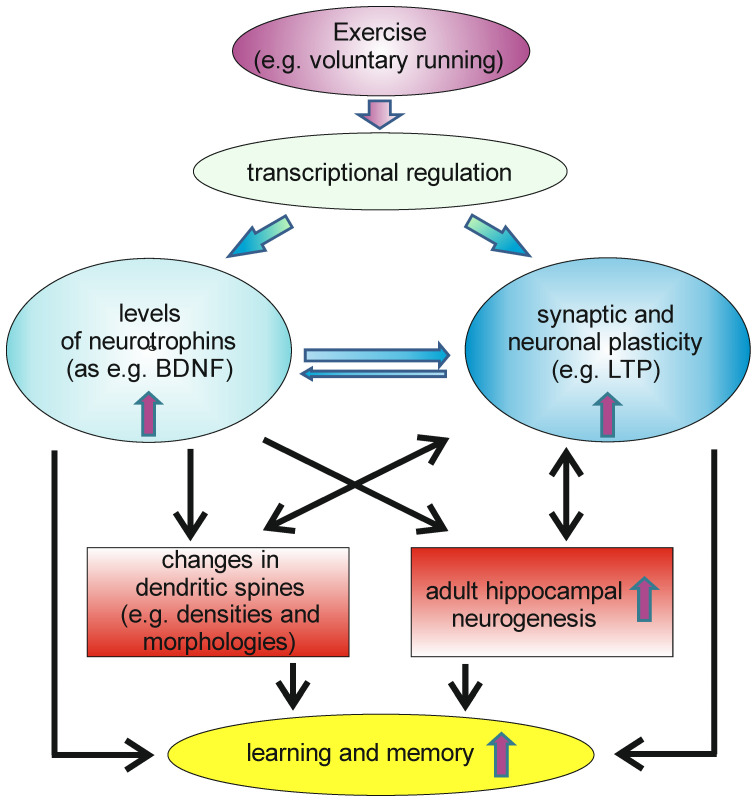
Voluntary exercise can induce transcription of different genes in neurons within the CNS, leading to increased levels of neurotrophins in the brain, as well as to enhanced synaptic and neuronal plasticity. This can, e.g., affect neuroanatomical correlates of learning and memory as, e.g., changes in dendritic spines, as well as increases in the rate of adult hippocampal neurogenesis. This, as well other changes induced by increases in neurotrophin levels and in synaptic and neuronal plasticity, have beneficial effects on learning and memory.

**Figure 3 biology-13-00558-f003:**
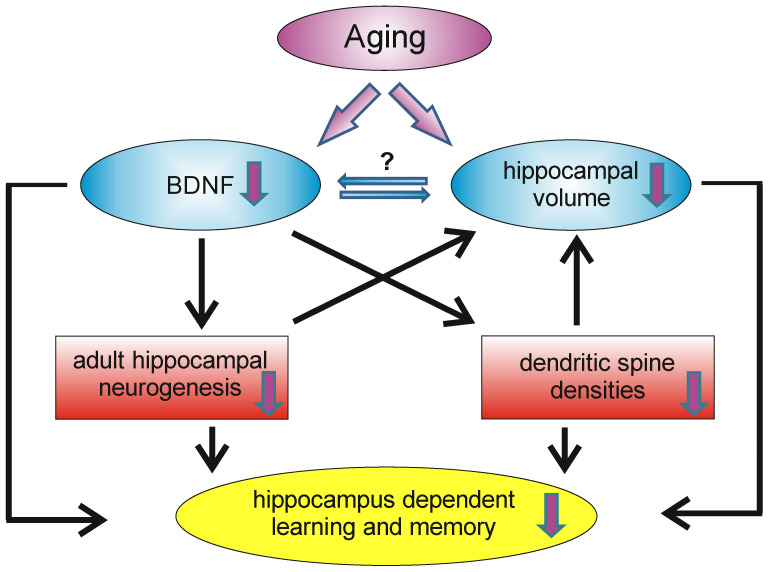
During aging, the hippocampus undergoes remodeling, which has an impact on learning and memory capacity. A decrease in BDNF levels can be observed, as well as volume reductions of the hippocampus. Decreased BDNF levels may influence the density of dendritic spines and the rate of adult hippocampal neurogenesis. Reductions in these two systems may contribute (on a morphological level) to the age-dependent volume reduction and may also contribute (among other factors) to the decrease in hippocampal learning and memory reductions.

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
