# Peer review of "The Neurotrophin System in the Postnatal Brain—An Introduction"

_biology, 2024, doi:10.3390/biology13080558_

Round 1

Reviewer 1 Report

Comments and Suggestions for Authors

This is a very well written and comprehensive review article that elegantly summarizes decades of work examining the roles of neurotrophins, and in particular BDNF, in the postnatal brain. This introduction to a special issue on neurotrophins gives a balanced view of the history behind the discovery of neurotrophins, early work on their role in neural plasticity, and more recent work on the association between BDNF polymorphisms with mental illness and eating disorders, as well as their positive roles in memory and aging through physical activity. The review cites important contribution in the field made using both animal models and human studies. This introductory review should be of significant interest to a broad audience in neuroscience research, as well as an updated and relevant summary for those more directly engaged in neurotrophic factor research.

Author Response

Thank you very much

Reviewer 2 Report

Comments and Suggestions for Authors This review describes about the role of neurotrophin in post natal brains.

Major comments

I found a recent review published on the topic at https://www.frontiersin.org/journals/molecular-neuroscience/articles/10.3389/fnmol.2023.1181397/full

Also many more reviews and works are already published on this topic which put me into trouble to recommend this paper for acceptance.

https://www.google.com/search?q=The+neurotrophin+system+in+the+postnatal+brain&rlz=1C1RLNS_enIN923IN923&oq=The+neurotrophin+system+in+the+postnatal+brain&aqs=chrome..69i57j33i160l3.504j0j7&sourceid=chrome&ie=UTF-8

So, how the authors justIfy about the novelty of this study.

Secondly, I don’t find the hypothesis of writing this review.  When reviews are available on the topic, the hypothesis must be very solid before writing the review.

Thirdly The introduction is very largely written. It needs drastic shortening to 3-4 para with 5-8 lines in each.

The last paragraph must state the hypothesis based on a clearly mentioned gap.

Minor

Although the title states that the review is on postnatal brain, they explain some other topics such as age related stuffs.

Author Response

The reviewer mentioned that this manuscript is not different from others recently published as e.g. the manuscript that can be foud  using this link: https://www.frontiersin.org/journals/molecular-neuroscience/articles/10.3389/fnmol.2023.1181397/full.
Indeed this is an review article (entitled:: “Building better brains: the pleiotropic function of neurotrophic factors in postnatal cerebellar development”). However the review is on the role of neurotrophic factors (including ephrins, EGF, GDNF and BDNF as well as other neurotrophic factor) on the cerebellum. In detail, this review focusses e.g. on the neurotrophic factors and cerebellar-related neurodevelopmental disorders. 
The other link the reviewer provided:https://www.google.com/search?q=The+neurotrophin+system+in+the+postnatal+brain&rlz=1C1RLNS_enIN923IN923&oq=The+neurotrophin+system+in+the+postnatal+brain&aqs=chrome..69i57j33i160l3.504j0j7&sourceid=chrome&ie=UTF-8 is just a google search for the terms 
“ The neurotrophin system in the postnatal brain”
 With the following results:

The “best” hit is linked to the invitation letter of the Special issue “Roles and Functions and their receptors in the brain” This review is written especially for this Special issue as an introducing article.

Concerning the novelty:
It is correct that we have only included some recent articles in the original version of the manuscript. 
This has been changes in the revised version and (see attached pdf file: (rev1-marked.pdf; the changes in the manuscript are highlighted in red).

The reviewer mentioned that: “although the title states that the review is on postnatal brain, they explain some other topics such as age related stuffs.”
Indeed, the review also addresses “age related stuff”. Aging is an unavoidable consequence of life and in the time course of aging, the postnatal brain is in one hand capable of adapting, but in the other hand there are age-related alterations that lead to changes in wiring of the postnatal brain structures leasing to changes in cognition, memorizing and learning. 
Likewise, the neurotrophin-system undergoes changes during postnatal life (e.g. the wiring of the hippocampus undergoes changes during childhood or there are changes in the aged (postnatal) brain as compared to the normal adult (postnatal) brain. Several of these changes are accompanied by changes in neurotrophins and/or their receptors).

Reviewer 3 Report

Comments and Suggestions for Authors

A lot of articles, including reviews, are devoted to neurotrophic factors. This article does not bring anything new to this topic. The review is illustrated with only one very general figure. New mechanisms of neurotrophins action that have been studied in the last at least 5 years are not discussed in the article. At the beginning of the review 1. Neurotrophic factors and neurotrophins, neurotrophins are indicated, but the authors pay attention to BDNF, one of the most studied neurotrophins, avoiding other factors regarding which there is conflicting and controversial data

Author Response

The review should serve as an introduction into the neurotrophin system in the postnatal brain. Indeed, the focus is mainly on BDNF. The reason for that is that most studies investigating postnatal brain (functions) are on BDNF.

Thus, it was not the intension to “avoid” the other neurotrophins.

Therefore, we tried also to include data on the functions of the other neurotrophins in the postnatal brain. Studies on the other neurotrophins in relation to the postnatal brain are rare, due to the fact that the expression of the other neurotrophins in the postnatal brain is not very prominent. In addition, the expression of the receptors trkA or p75NTR is also very low in the postnatal brain and restricted to some brain areas (during brain development, the expression pattern of the neurotrophins and their cognate receptors in extremely different as compared to the postnatal situation). We also added this to the manuscript since this might be important for the interpretation of some recent published data.

The reviewer noted that only some new publications on the neurotrophin system in the postnatal brain are mentioned in the manuscript. This is correct. Therefore, in order to bring the manuscript more up-to-date, we have added several paragraphs to the manuscript dealing with recent data on the neurotrophin system in the postnatal brain. In summary about 25 citations were added to the revised manuscript.

Changes in the manuscript are highlighted in red and can be seen in the attached pdf-file (Rev1-marked.pdf)

Round 2

Reviewer 2 Report

Comments and Suggestions for Authors

The ms improved. 

Author Response

Thank you

Reviewer 3 Report

Comments and Suggestions for Authors

I agree that most research in the postnatal brain focuses on BDNF, and there are quite a few reviews about it. Thus, the relevance of the presented review is questionable. As for other neurotrophins, the search reveals the presence of modern experimental work that the authors could analyze. In the presented review, I do not see any drawings or diagrams that could be interesting or relevant, and in general the article is poorly illustrated.

Author Response

The reviewer mentioned that the review article is poorly illustrated. Based on that, we added two further illustrations to the manuscript.(figure 2 and figure 3 in the current version of the manuscript).

We hope that the manuscript is now better illustrated.

The changes (e.g. figure legends) are highlighted in red in the marked version of the revised manuscript.
